# Learning to Follow Object-Centric Image Editing Instructions Faithfully

**Tuhin Chakrabarty**[1*]   **Kanishk Singh**[1*]   **Arkadiy Saakyan**[1]   **Smaranda Muresan**[1,2]

[1]Department of Computer Science, Columbia University
[2]Data Science Institute, Columbia University

tuhin.chakr@cs.columbia.edu, ks4038@columbia.edu, a.saakyan@cs.columbia.edu, smara@columbia.edu

## Abstract

Natural language instructions are a powerful interface for editing the outputs of text-to-image diffusion models. However, several challenges need to be addressed: 1) underspecification (the need to model the implicit meaning of instructions) 2) grounding (the need to localize where the edit has to be performed), 3) faithfulness (the need to preserve the elements of the image not affected by the edit instruction). Current approaches focusing on image editing with natural language instructions rely on automatically generated paired data, which, as shown in our investigation, is noisy and sometimes nonsensical, exacerbating the above issues. Building on recent advances in segmentation, Chain-of-Thought prompting, and visual question answering, we significantly improve the quality of the paired data. In addition, we enhance the supervision signal by highlighting parts of the image that need to be changed by the instruction. The model fine-tuned on the improved data is capable of performing fine-grained object-centric edits better than state-of-the-art baselines, mitigating the problems outlined above, as shown by automatic and human evaluations. Moreover, our model is capable of generalizing to domains unseen during training, such as visual metaphors.

## 1 Introduction

Frameworks for large-scale text-conditional image synthesis, which rely on diffusion processes (Saharia et al., 2022; Rombach et al., 2022a; Ramesh et al., 2021; Nichol et al., 2021) have shown impressive generative capabilities and practical uses. Notably, image editing guided by text has garnered considerable attention due to its ease of use and seemingly high-quality results (Avrahami et al., 2022b,a; Hertz et al., 2022a; Kawar et al., 2023). These advances are now utilized in leading industry

---

* Equal Contribution

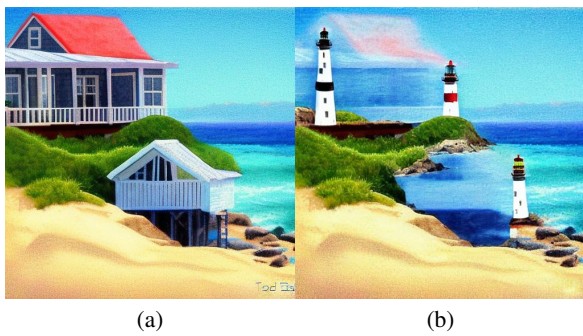

(a)                                  (b)

Figure 1: (a) Input Image (b) Edited image from InstructPix2Pix (Brooks et al., 2023) with the instruction *Add a lighthouse.*

tools such as Adobe Photoshop[1], bridging the gap between technology and content creators.

While natural language instructions act as a powerful interface for editing images, following them remains a challenging task for several reasons. First, these instructions are often *underspecified*, requiring models to uncover their implicit meaning. For example, in Figure 1 for the input image (a), the user provides a prompt *Add a lighthouse*. The model needs to understand how a lighthouse looks, that only one lighthouse needs to be added and that it needs to be placed on land and not in the water. Second, models must be able to localize where the "background" is in the image so that the lighthouse can be added appropriately (*grounding*). Finally, models must follow instructions *faithfully*, i.e. preserve the elements of the image not affected by the edit instruction (e.g., both houses in Figure 1 (a)).

One of the key challenges for making progress on the task of image editing via natural language instructions is the lack of high-quality annotated or naturally occurring data. Recently, Brooks et al. (2023) proposed a way to handle this by automatically creating a paired dataset utilizing large language models and text-to-image models. They

---

[1]adobe.com/sensei/generative-ai

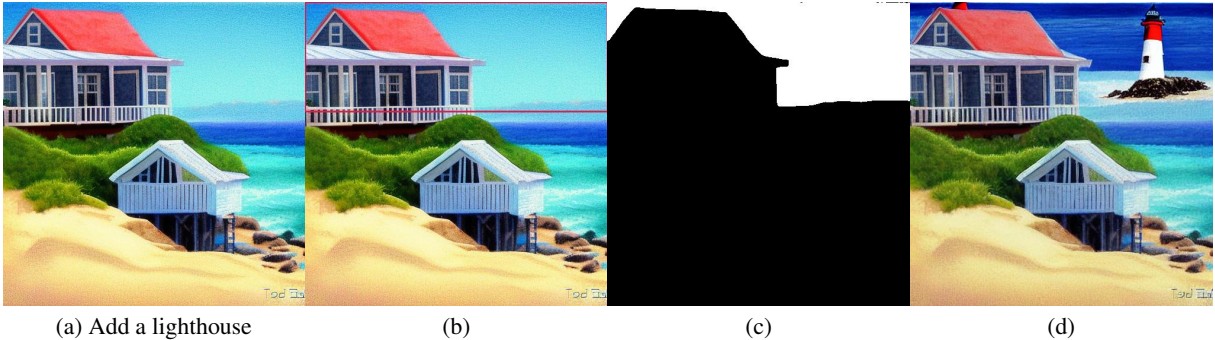

| (a) Add a lighthouse | (b) | (c) | (d) |

Figure 2: Steps to create parallel data: (a) Input Image + Edit Instruction; (b) Image with grounding in the form of a bounding box for the entity where transformation has to be made; (c) Masked localized segment of the grounded image where the transformation has to be made; (d) Final output.

further train a conditional diffusion model on this paired data of synthetic examples and show that their model is capable of editing images conditioned on natural language instructions at run-time. However, closer inspection reveals that the data is noisy and sometimes nonsensical. As can be seen in Figure 1 (b) the model adds three lighthouses instead of one, likely due to the underspecification of the instruction and the lack of grounding. Furthermore, it compromises faithfulness by removing both houses from the input image.

To tackle these challenges we first create a high-quality corpus starting from the parallel data released by Brooks et al. (2023). For example, given the caption of the input image in Figure 1, *Ocean Cottage by Todd Baxter*, and an edit instruction *Add a lighthouse*, we use recent advances in Chain-of-Thought prompting (Wei et al., 2022) to identify whether the transformation can be performed in the context of the input image and what entity should be transformed. The entity and the input image are fed to an object detection model GroundingDINO (Liu et al., 2023). As can be seen in Figure 2 (b), DINO draws a bounding box to denote "background". However, grounding in the form of bounding boxes cannot entirely disentangle the entity of interest. Thus, we use recent advances in image segmentation (Kirillov et al., 2023) to identify the exact segment containing the entity that needs to be masked (see Figure 2 (c)) and and use Stable Diffusion+Inpainting (Rombach et al., 2022a) with the masked image (c) to obtain the edited output (Figure 2(d)).

To further account for faithfulness in our paired training data we leverage techniques from VQA (Antol et al., 2015) to ensure that the edited im-

ages remain faithful to the entities in the original image. For this, we generate relevant questions with 'Yes/No' answers regarding the unmodified elements in the image such as "*Does this image contain a cottage*" or questions regarding objects in the instruction "*Is there a lighthouse in the image*" using the Vicuna-13B model (Chiang et al., 2023). Given an image-question pair, we then use the BLIP-2 VQA-model Li et al. (2023) to collect responses and re-rank generated images from Stable Diffusion+Inpainting by faithfulness scores in terms of correct answers and select the best one for higher quality. We then fine-tune the model on this newly created parallel data. In addition, both during finetuning and inference, we enhance the supervision signal by denoising parts of the image that need to be changed by the instruction as shown in Figure 5.

To summarize, our contributions are:

- Improving the quality of existing paired datasets used for image editing via natural language instructions with the help of recent advances in segmentation, Chain-of-Thought prompting, and visual question answering.

- Curating a test set of diverse non-noisy instructions, consisting of both in-domain and out-of-domain examples and conducting a thorough evaluation across SOTA baselines and our model ablations.

- Demonstrating that fine-tuning a diffusion model on our parallel data enhanced with supervision signal leads to a significant improvement over several compelling baselines in terms of faithfulness using TIFA scores (Hu et al., 2023) as well as instruction satisfiability. Our human evaluation corroborates these findings.

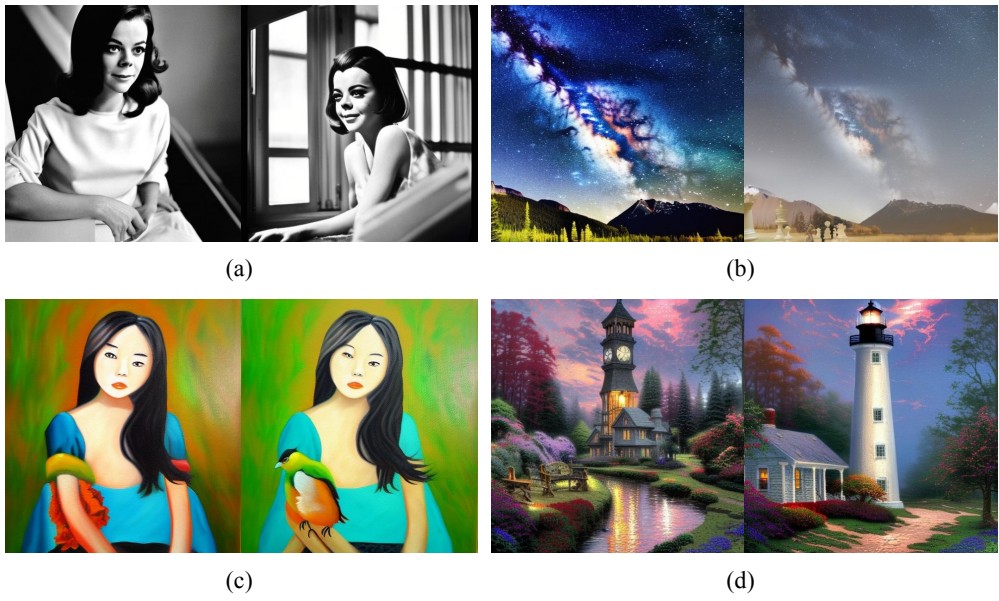

(a)               (b)

(c)               (d)

Figure 3: Examples of noisy edit instructions and image-pairs: (a) *Make her look like an android*; (b) *Make the rocky mountains look like a chessboard*; (c) *Replace her with a bird*; (d) *Have it be a lighthouse*.

We release our code, data and pretrained models.[2]

## 2 Data

### 2.1 Problems with Existing Datasets

The dataset introduced by (Brooks et al., 2023) marked a significant step towards enabling diffusion models to comprehend instructions for image editing. However, since GPT-3 was used to generate captions and edit instructions (Figure 9 in Appendix A), the constructed image-edit pairs suffered from various limitations that makes the editing process less precise and efficient.

One of the frequent concerns in the edit instructions is that many of them are vague and incomprehensible to perform a successful edit. Furthermore, many of the edit instructions suffer from language model hallucinations. For instance, in Figure 3 (b), the instruction *"Make the rocky mountains look like a chessboard"* is not sensible in the context of the input image. In Figure 3 (c), the gold image does not actually represent the instruction, creating issues in the instruction following the fine-tuned model. Furthermore, object-centric instructions are inherently harder for diffusion models. Models that cannot localize the corresponding entity or region in the image end up performing incorrect or excessive modifications, resulting in images that are not faithful to the instruction or input image as can be seen in Figures 3 (c) and (d). This undermines the

quality of the training set and leads to non-faithful edits by fine-tuned diffusion models.

### 2.2 High-Quality Training Dataset Curation

We describe our approach for curating the dataset with focus on addressing the above-mentioned challenges including underspecification, grounding, and faithfulness.

**Filtering noise and handling under-specification** To improve the noisy synthetically generated edit instructions from (Brooks et al., 2023), we leverage the reasoning capabilities of large language models through Chain-of-Thought prompting (Wei et al., 2022) to jointly predict whether the instruction is appropriate w.r.t the context present in the original caption and to generate the entity/region that needs to be grounded and changed for performing the edit. Table 3 in Appendix A shows how we jointly elicit the edit entity as well as a verdict on whether the transformation is possible. Large language models can uncover implicit entities in image editing tasks based on their own commonsense knowledge, even without explicit mentions in the input caption or edit instruction. For instance, in Figure 4 given the original image with the caption *Buttermere Lake District* and edit instruction *Add an aurora borealis*, the model outputs that the entity to which the edit is applied is *sky*, using the implicit commonsense knowledge that "aurora borealis" has to appear in the sky.

We filter object-centric image-instruction pairs

[2]github.com/tuhinjubcse/FaithfulEdits_EMNLP2023

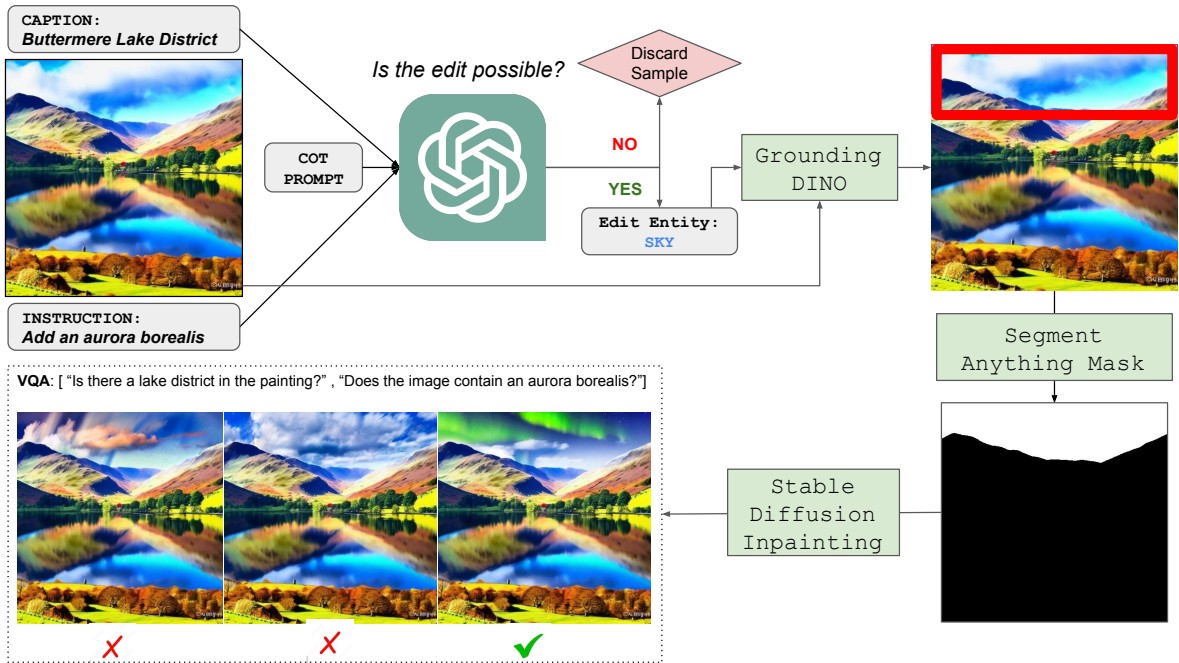

Figure 4: Steps to create high-quality training parallel data: Given an input image, caption, and edit instruction we first use Chain-of-Thought (CoT) prompting with ChatGPT to identify whether the edit instruction is sensible and if it is, what is the entity that needs to be transformed. Using the LLM-generated edit entity we use GroundingDINO to localize it and SAM (Segment Anything Mask) to segment it. We then use Stable Diffusion Inpainting to generate 3 images and filter out the best image with the help of VQA.

that align with the original caption, discarding images lacking a specific segment for editing. This ensures high-quality, precise, and contextually appropriate entity-centric image editing.

**Incorporating Grounding** We utilize object detection and segmentation to provide additional supervision signals for grounded-image editing. The edit entity generated from ChatGPT (gpt-3.5-turbo) in the previous step is given as an input to the open-set object detection model GroundingDINO (Liu et al., 2023) that generates a rectangular box around the region/entity that needs to be altered. After getting the box coordinates from GroundingDINO, we further perform the image segmentation using the SOTA model SAM (Segment Anything Mask) (Kirillov et al., 2023), which takes point inputs for generating a segmentation mask over the entity. These steps can be seen in Figure 4: ChatGPT outputs the entity to which the edit has to be applied given the caption and the instruction (*sky*), GroundingDINO locates the entity (red rectangular box), and the SAM model further disentangles the sky from the mountains.

**Generating Images using Stable Diffusion Inpainting** Stable Diffusion Inpainting is a latent text-to-image diffusion model capable of generating photo-realistic images given any text input, with the extra capability of inpainting a specified area of an image given a mask. However, this model does not understand implicit instructions and requires captions of the original image, so we use the "edited caption" (See Figure 9 in Appendix A) present in the existing dataset to perform the edit operation on the image. We feed the binary image-segmentation mask generated by the SAM model as seen in Figure 4 along with the input image and the edited caption to perform the required entity-centric edit on the image. Stable Diffusion Inpainting generated images might not be always faithful, so we generate three images per input and re-rank them based on faithfulness as described below.

**Ensuring Faithfulness** To further account for faithfulness, we leverage techniques from Visual Question Answering (Antol et al., 2015) to ensure that the edited images are faithful to the instruction as well as to the entities in the original image. We formulate relevant questions regarding the unmodified elements in the image such as the presence of specific objects or contextual information. We gen-

erate these questions using the Vicuna-13B model (Chiang et al., 2023), a LLaMA model (Touvron et al., 2023) fine-tuned for instruction following. We use the edited caption available in the dataset to extract noun phrases and drop entities which are either locations or names of individuals. For example, in Figure 4, given the edited caption *"Buttermere Lake District with Aurora Borealis"* and the extracted entities *Buttermere*, *Lake District* and *Aurora Borealis*, we drop Buttermere and pass the other two entities to the Vicuna model to generate question-answer pairs. Figure 4 shows the generated questions corresponding to the individual entities.

For *Remove* or *Delete* instructions, we need to ensure that the edit entity is *not* present in the resulting image. Along with the edited caption and the extracted entities, we also pass the edit instruction to generate a question ensuring a successful edit. For instance, suppose the original caption is *Buttermere Lake District with Aurora Borealis* and the edit instruction is *Remove Aurora Borealis*. We input the edited caption *Buttermere Lake District*, the entity *lake district*, and the edit instruction to generate the following questions: *Is there a lake district in the picture?* and *Does the picture contain Aurora Borealis?* along with the corresponding correct answers *Yes*, *No*.

We input the questions and the image generated by Stable Diffusion Inpainting in the SOTA VQA-model BLIP-2 (Li et al., 2023) to extract whether entities are present in the edited image. We count the number of correct answers (*No* for *remove* instructions, *Yes* for other instructions) for each of the three images generated using Stable Diffusion Inpainting and select the one having the largest number of correct answers.

### 2.3 Test Data creation

Our test set of 465 <image, instruction> pairs is curated carefully across both in-domain and out-of-domain images to account for model robustness and generalization capabilities. To create an in-domain test set, we deployed a filtering strategy where we only retained those examples in our test having a CLIP-similarity score (Radford et al., 2021) less than 0.7 with every other training image. We further focus on multiple carefully chosen verbs such as *Replace, Swap, Add, Turn, Change* and generate 20 images per verb leading to 200 diverse <image, instruction> pairs.

| Dataset Type | Total Pairs |
|---|---|
| Instr-Pix2Pix (In-domain) | 200 |
| Magic-Brush (Out-domain) | 200 |
| Visual-Metaphors (Out-domain) | 65 |

Table 1: Test split and number of image-caption pairs.

For creating an out-of-domain test set, we use two sources of examples. First, we consider the recently released MagicBrush dataset (Zhang et al., 2023). This is a manually annotated dataset for instruction-guided image editing that covers multiple scenarios including single-turn and multi-turn edits. They sample real-world images from the MS-COCO (Lin et al., 2015) dataset, ask annotators to write instructions, and use the DALLE-2 (Ramesh et al., 2022) image editing platform to interactively synthesize target images. For our experiments, we only consider single-turn edits. We discard the *Change Action* instructions such as *Make the person jump*, *Make the dog look away* as they were not present in the original InstructPix2Pix data. We finally select 200 random images from the MagicBrush test set after the above considerations.

Second, we use a dataset of 100 DALLE-generated imperfect visual metaphor images paired with expert-written natural language instructions to improve them (Chakrabarty et al., 2023). We select 65 single-turn images with verbs that are already present in our in-domain test set.

## 3 Fine-tuning with Additional Supervision

Our training set curation pipeline led to a set of 52,208 high-quality instruction-images pairs. We split this data in 80:10:10 for the training, validation, and test sets, respectively.

We follow the same protocol as InstructPix2Pix training and use their codebase. For an image x, the diffusion process adds noise to the encoded latent producing a noisy latent $z = E(x)$ where the noise level $z_t$ increases over timesteps $t \in T$. We learn a network $\theta$ that predicts the noise added to the noisy latent $z_t$ given image conditioning $c_I$ and text instruction conditioning $c_T$. We minimize the latent diffusion objective and initialize the weights of our model with the InstructPix2Pix checkpoint. To support image conditioning, we add input channels to the first convolutional layer, concatenating $z_t$ and $E(c_I)$. All available weights of the diffusion model are initialized from the pre-trained checkpoints, and weights that operate on the newly added input channels are initialized to zero. We reuse the

same text conditioning mechanism that was originally intended for captions to instead take as input the text edit instruction $c_T$. We experiment with two different training strategies with additional supervision on top of the vanilla InstructPix2Pix.

1. **Bounding Box Supervision**: During fine-tuning, the image conditioning $c_I$ is set to an image with the bounding box around the region of interest (see Figure 5 (a)).

2. **Segmentation Mask Supervision**: During fine-tuning, the image conditioning $c_I$ is set to the image with a denoised segmentation mask around the region/entity of interest. (see Figure 5 (b)).

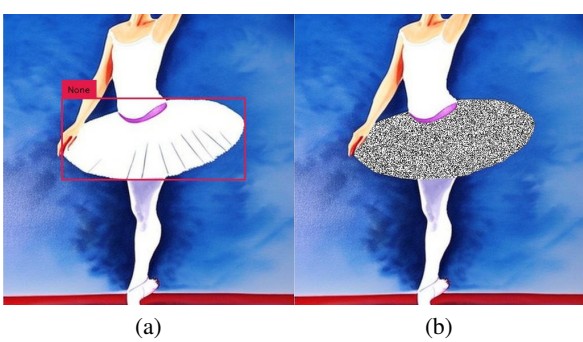

Figure 5: (a) Bounding Box supervision (b) Segmentation Mask supervision. Edit instruction: *make the skirt red*.

We initialized our model weights from the InstructPix2Pix model and finetuned for an additional 8k steps on NVIDIA-A100GPUs. We adopt the rest of the hyperparameters from the public InstructPix2Pix public repository. Further details for hyperparameters during inference are mentioned in Appendix A.

## 4 Models

Below we describe how we use some of the state-of-the-art baselines as well as ablations of our best model to generate outputs based on the test set instruction-image pairs.

**InstructPix2Pix** We use the off-the-shelf InstructPix2Pix (Brooks et al., 2023) model checkpoint.

**Instruct-X-Decoder** Zou et al. (2023) released the X-Decoder model that provides a unified way to support all types of image segmentation and a variety of vision-language (VL) tasks. Given the high-quality referring segmentation results with X-Decoder, they further combine it with an off-the-shelf Stable Diffusion image inpainting model and perform zero-shot referring image editing.[3]

**Grounded Inpainting** Grounded-Inpainting module proposed by (Liu et al., 2023) deployed with `gpt-3.5-turbo` language model to identify the region/object of interest from the corresponding edit instruction. The pipeline then includes identifying the bounding box for the region of interest and generating a binary segmentation mask before passing it to the Stable Diffusion Inpainting model (Rombach et al., 2022b). We only provide the instruction to the inpainting module as opposed to the complete caption during the training dataset curation for fairness to other baselines.

**InstructPix2Pix+BoundingBox** We use the model fine-tuned with a bounding box supervision signal described in section 3. During the inference, the bounding box is constructed with the GroundingDINO model (Liu et al., 2023) using the entity extracted from the edit instruction by `gpt-3.5-turbo`.

**InstructPix2Pix+EntityMask** We use the model fine-tuned with a segmentation mask supervision signal described in section 3. During the inference, the segmentation mask is constructed with the Segment Anything Mask (SAM) model (Kirillov et al., 2023) using the entity extracted from the edit instruction by `gpt-3.5-turbo`.

## 5 Evaluation

**Automatic Evaluation using TIFA-Score** We use a recent metric proposed by Hu et al. (2023), TIFA (Text-to-image Faithfulness evaluation with question-answering),[4] for evaluating the faithfulness of the generated images to their text inputs. TIFA evaluates a generated image using a two-stage pipeline: first, generates a list of question-answer pairs that cover various aspects of the contextual information provided in the given caption, and then uses SOTA VQA models such as BLIP-2 to answer these questions and match the correct answers for faithfulness. This framework is proposed for benchmarking diffusion models. The framework generates 7-10 question-answer pairs per image

---

[3] `github.com/microsoft/X-Decoder`
[4] `tifa-benchmark.github.io`

caption (modified, which covers the instructional edit aspects too). We average the score for individual images over all these question pairs. The final score is reported as the average of the TIFA score over all the images in the test set.

**Human evaluation** We randomly sample 100 examples from our test set with 50 images from in-domain and 50 images from out-of-domain. We recruit 31 annotators on Amazon Mechanical Turk through a rigorous qualification test. We require three distinct workers to do one HIT. Given an input image and an edit instruction, they are asked to judge if the output images from five systems satisfy the edit by choosing between *Yes*, *Partially Yes*, and *No*. Following prior work (Kayser et al., 2021; Majumder et al., 2021; Chakrabarty et al., 2022), we map these to $1, \frac{1}{2}, 0$ respectively and report the average as $H_{score}$. Additionally, we ask the annotators to consider how faithful the output image is to the input: if the output image changes several elements that were beyond the scope of the edit instruction, they are asked to respond *No*. We require them to provide justification for their choice (see Figure 6) to prevent random guessing.

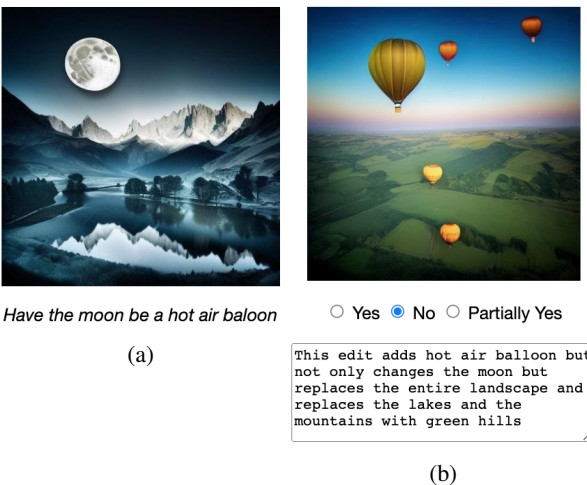

*Have the moon be a hot air baloon*

(a)

○ Yes  ● No  ○ Partially Yes

```
This edit adds hot air balloon but
not only changes the moon but
replaces the entire landscape and
replaces the lakes and the
mountains with green hills
```

(b)

Figure 6: Human evaluation framework (a) Input Image with Edit Instruction (b) Example of a bad edit

## 6 Results

Table 2 shows the performance of our model and the baselines on the in-domain and out-of-domain parts of the test set in terms of both automatic and human evaluation. *InstructPix2Pix+EntityMask* appears to be the winning system with *Instruct-Pix2Pix+BoundingBox* coming second. These results indicate that our improved data combined with the additional supervision signal lead to better edits.

| System | TIFA | $H_{score}$-I | $H_{score}$-O |
|---|---|---|---|
| Instruct-X-Decoder | 58.10 | 29.1 | 17.5 |
| Grounded Inpainting | 56.74 | 40.3 | 22.3 |
| InstrP2P | 62.24 | 49.3 | 25.2 |
| InstrP2P+BOUNDINGBOX | 63.39 | 62.8 | 37.7 |
| InstrP2P+ENTITYMASK | **65.84** | **69.8** | **65.1** |

Table 2: Automatic evaluation using TIFA scores on the entire test set and human evaluation on in-domain($H_{score}$-I) and out-of-domain ($H_{score}$-O) parts of the test set.

TIFA acts well as a reference-free automatic evaluation metric and is useful in real-world settings gold images are not necessarily available. For instance, given the edited caption *Red skirt ballerina* for the edit *Make it a red skirt* (see top row in Figure 7), TIFA generates several <question, choices, answer> tuples such as <Is there a red skirt?, [Yes, No], Yes>, <Who is wearing the red skirt?, [Ballerina, Singer, Actress, Model], Ballerina> , <What color is the skirt?, [Red, Blue, Green, Yellow], Red> which are then scored by a VQA model for correctness.

For human evaluation, as mentioned earlier, three MTurkers were recruited for each instance. The IAA using Krippendorff's $\alpha$ (Krippendorff, 2011) is 0.58, indicating moderate agreement. Human judges consistently preferred our *InstructPix2Pix+Entity Mask* model and the gap between our best model and the other baselines are substantial. All judges unanimously agreed that the original InstructPix2Pix changes the images following the instructions; however, the resulting images exhibit excessive modification and lack photorealism. For instance, for image (iv) in the top row in Figure 7 human judges rate the edit as bad and give the following justification: *The entire dress has been changed to red and the background is a different color*. These explanations shed light on user preferences corroborating the fact that humans like the edits to be precise and faithful. Qualitative examples in Figure 8 demonstrate that our model is capable of generalizing to instructions and images (primarily illustrations) from a completely different domain. See more examples in Appendix B.

## 7 Related Work

Prior work can be categorized primarily among text-guided image editing, involving the application of multiple models such as language mod-

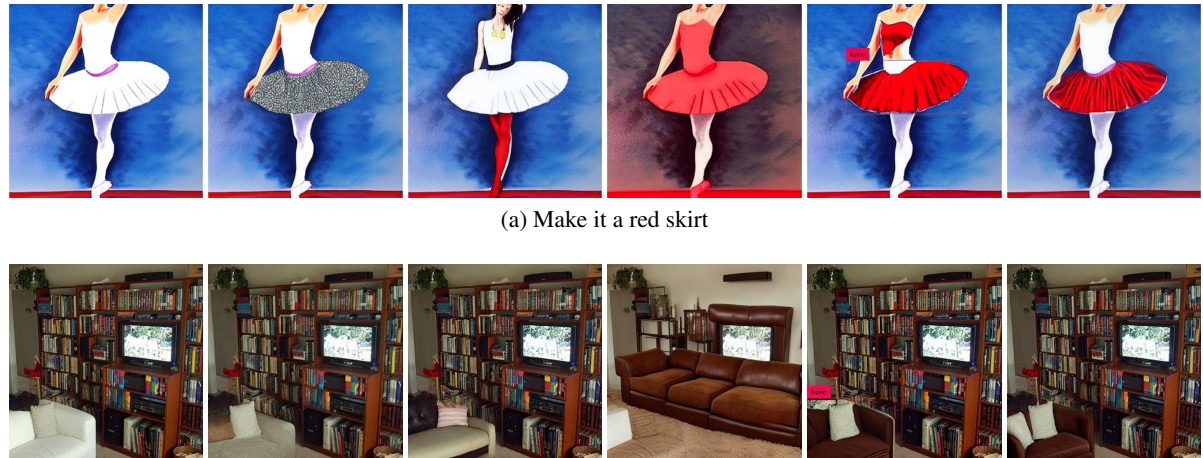

(a) Make it a red skirt

(b) Change the white couch to a brown couch

Figure 7: Sample generations for an image from (a) instruct-pix2pix dataset (b) Magicbrush dataset. The images displayed are following order (i) input-image, (ii) Grounded-Inpainting, (iii) X-Decoder, (iv) InstructPix2Pix, (v) InstructPix2Pix+BoundingBox, (vi) InstructPix2Pix+EntityMask

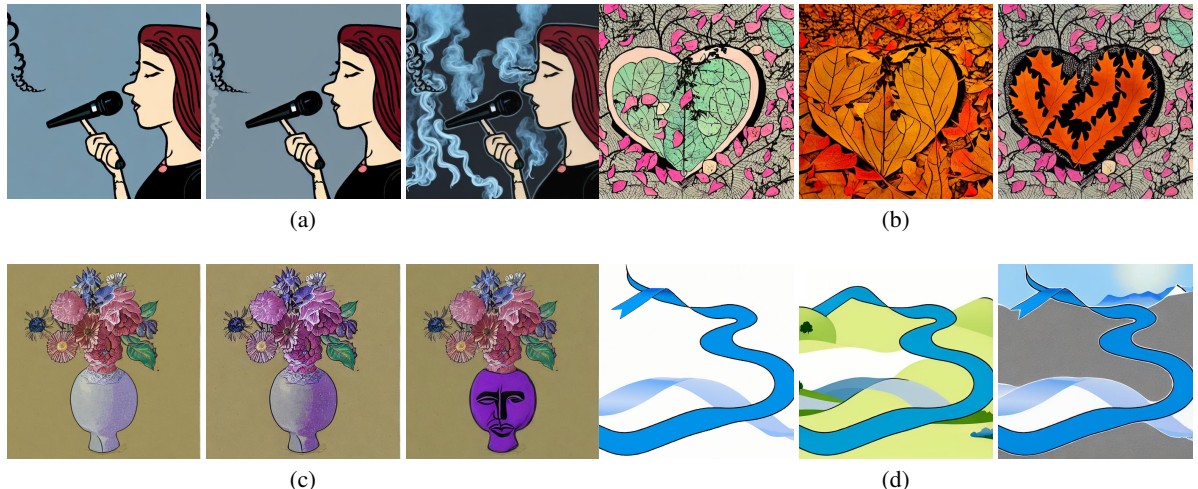

(a)                                                    (b)

(c)                                                    (d)

Figure 8: Sample edits showing <Input, InstructPix2Pix, InstructPix2Pix+Entity Mask> for the metaphors and corresponding edit instructions (a)*["Her song is like a cloud of heavy smoke", "Add more smoke near the microphone"]* (b)*["My heart is a garden tired with autumn", "Change the heart such that it is made of autumn leaves"]* (c) *["My head is like a vase growing","Replace the body of the purple vase with the face of a man"]* (d)*["The river is a ribbon wide", "Add a landscape"]*

els and diffusion models in a pipelined fashion and purely diffusion-based generative models. Approaches falling under the paradigm of text-guided image editing can be further subdivided into global-description-guided editing and local-description-guided editing. Imagic (Kawar et al., 2023) refines a textual representation so that it aligns with a given image and then blends this with a target description to create diverse image edits. On the other hand, Text2LIVE (Bar-Tal et al., 2022) employs a unique approach that teaches a model to generate an edit layer. This layer is then merged with the input image, allowing for localized modifications. Coua-

iron et al. (2022) automatically generates a mask highlighting regions of the input image that need to be edited, by contrasting predictions of a diffusion model conditioned on different text prompts. They further rely on latent inference to preserve content in those regions of interest and show excellent synergies with mask-based diffusion.

Methods such as Prompt2Prompt Hertz et al. (2022b) utilize both local and global parts of the image using a cross-attention network to perform edits. Approaches proposed by Avrahami et al. (2022b); Wang et al. (2023) work on regional descriptions for localized editing. Instruction-guided

editing as suggested by Brooks et al. (2023); Chen et al. (2023); Fu et al. (2020); El-Nouby et al. (2019) argues towards an approach to edit images via language instructions without explicitly mentioning the contextual information. Recent works from Liu et al. (2023) explore a strong object-detection model coupled with strong segmentation (Kirillov et al., 2023) model to edit via Stable Diffusion model (Rombach et al., 2022a). Further work (Saharia et al., 2022), (Richardson et al., 2021), (Fu et al., 2020) explores stylistic-image editing including via generative models. Unlike several existing prior works we focus on the faithfulness and specificity of object-centric edits. Like Zhang et al. (2023) we argue that high-quality training data and incorporating grounding is the key to achieving high-quality edits.

Recently there has been a growing interest in using text-to-image diffusion models for creative tasks such as creating illustrations or abstract art. Akula et al. (2023) release MetaCLUE, consisting of four interesting tasks (Classification, Understanding, Localization, and Generation) related to metaphorical interpretation and generation of images. Chakrabarty et al. (2023) release a dataset of visual metaphors through collaboration between large language models and text-to-image models. These model-generated outputs while being impressive are often imperfect and require further edits. Our results on editing imperfect visual metaphors open up opportunities for content creators who can simply use natural language instructions to steer AI-generated images to their liking.

## 8 Conclusion

We address challenges in natural language-based image editing tasks and provide a novel approach to enhance the quality of the training data. We improve the supervision signal and tackle the issues of underspecification, grounding, and faithfulness by leveraging advancements in segmentation, Chain-of-Thought prompting, and VQA. Our models fine-tuned on the improved dataset with enhanced supervision signal outperform the existing baselines on object-centric image editing both in terms of automatic and human evaluation. Moreover, our models showcase the capability to edit faithfully on out-of-domain datasets. Overall our findings highlight the significance of high-quality annotation and grounded supervision signals for precise and faithful image editing.

## Limitations

While our best-performing model supports various edit types on real images, we do not benchmark for global editing (e.g., style transfer). Our method can greatly enhance text-guided image editing, making it accessible to more users without professional knowledge, boosting their efficiency. However, the risk of misuse for creating fake or harmful content is a concern. Therefore, implementing robust safeguards and responsible AI protocols is critical. Finally, while our data creation uses a pipeline of best-performing state-of-the-art models, there is still potential for error in our training data. Additionally, while TIFA score acts as a good reference-free metric for automatic evaluation it is relying on VQA model's correct answers which may be incorrect. Thus, we corroborate our results with human evaluation. Our study focuses on single-turn atomic instructions and does not show results on multi-turn instructional edits. Additionally, our method only works for natural language instructions in English and does not handle instructions in other languages.

## Ethics Statement

The use of text-to-image generation models is subject to concerns about intellectual property and copyright of the images generated since the models are trained on web-crawled images. We use the LAION-Aesthetics dataset which primarily consists of images from a variety of mediums (photographs, paintings, digital artwork). To ensure the collection of high-quality human annotations and fair treatment of our crowd workers, we have implemented a meticulous payment plan for the AMT task. We conduct a pilot study to estimate the average time required to complete a session. We pay our workers 18$/hr, which is above minimum wage. All data collected by human respondents were anonymized and only pertained to the data they were being shown. We do not report demographic or geographic information so as to maintain full anonymity. Workers were paid their wages in full immediately upon the completion of their work.

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

## A  Pormpts, hyperparameters, annotation

### A.1  Few-shot Prompts for filtering noise and handling under-specification

We describe our few-shot prompts given to the ChatGPT model (`gpt-3.5-turbo`) for our dataset-generation pipeline.

| |
|---|
| You will be given an input caption of an image and an instruction to transform it by an image editor. Sometimes, however, the instruction does not make sense as the resulting transformation would result in a nonsensical image. Based on the input caption and instruction, reason how the resulting image would look like and whether the resulting image would be possible to imagine. Provide your verdict on whether the transformation is possible. If the verdict is true, also state the entity. First, provide your reasoning, starting with the words "*The resulting image would show...*". Then, return the verdict and the entity in JSON format. |
| Caption: Barn In Autumn Smoky Mountains by David Chasey 
 Instruction: Change the barn to a castle 
 *The resulting image would show a castle in the mountains, which is a sensible image.* 
 { "verdict": "true", 
 "entity": "barn"} |
| Caption: Sligachan Bridge by English Landscape Prints, 
 Instruction: change the bridge to a wooden ship, 
 *The resulting image would show a ship up in the air which does not make logical sense.* 
 { "verdict": "false", 
 "entity": "none"} |
| Caption:...... |
| Caption:...... |
| Caption:...... |

Table 3: Five shot prompt given to ChatGPT (`gpt-3.5-turbo`) to elicit verdict on whether instruction is valid in the context of input image caption and edit entity

### A.2  Few-shot Prompts for generating questions using Vicuna model

For question-answer pair generation to evaluate faithfulness during the dataset cutation, we provide three-shot prompt to the Vicuna model as provided in Table-4.

### A.3  Hyperparameters for training and inference

We finetuned the the InstructPix2Pix checkpoint from their official repository for $8k$ steps on NVIDIA-A100GPUs with a learning rate of $1e-4$. We take the rest of the hyper-parameters from the official implementation of InstructPix2Pix repository. For generation during inference, we set cfg-text=7.5 and cfg-image=1.5 for instruct-pix2pix

| |
|---|
| You are given an image description and the corresponding entities present in the caption. Generate a question per entity to check whether the image aligns with the text. |
| Caption: handsome man wearing a tuxedo and top hat in casual style clothes over blurred mountain background. Entity: Handsome man, tuxedo and top hat, casual style clothes, blurred mountain background 
 Entity: Handsome man 
 Question: Is the person in the picture a handsome man? 
 Answer: Yes 
 Entity: Tuxedo and top hat 
 Question: Is the man wearing a tuxedo and a top hat? 
 Answer: Yes 
 Entity: Casual style clothes 
 Question: Is the man dressed in casual style clothes? 
 Answer: Yes 
 Entity: Blurred mountain background 
 Question: Is there a blurred mountain background in the picture? 
 Answer: Yes |
| Caption:...... |
| Caption:...... |

Table 4: Three shot prompt given to Vicuna-13b to generate question for ensuring faithfulness wrt original caption and instruction

baselines and its ablations. For evaluation using TIFA, we use GPT-4 for generating questions and flan-t5-xxl version of the BLIP-2 model for visual question answering.

### A.4  InstructPix2Pix training data creation

Figure 9 shows the training data creation pipeline for InstructPix2Pix by (Brooks et al., 2023).

### A.5  Human annotation and explanations

Table 5 shows the rationale chosen by human judges for edits from different baselines. Image in row 1 + Entity Mask is deemed as perfect as can be seen in the written explanation. The vanilla model edits the image too much beyond the scope as can be seen in the images in row 3 and 4 of Table 5.

## B  Sample generations for the Metaphor and MagicBrush datasets

Figure 10 and 11 show sample image generations from the Metaphor dataset with associated metaphor-descriptions and corresponding edit instructions. Figure 12 and 13 show sample generations for the Magicbrush dataset. The images displayed are following order (i) input-image, (ii) Grounded-Inpainting, (iii) X-Decoder, (iv) InstructPix2Pix, (v) InstructPix2Pix+BoundingBox, (vi) InstructPix2Pix+EntityMask

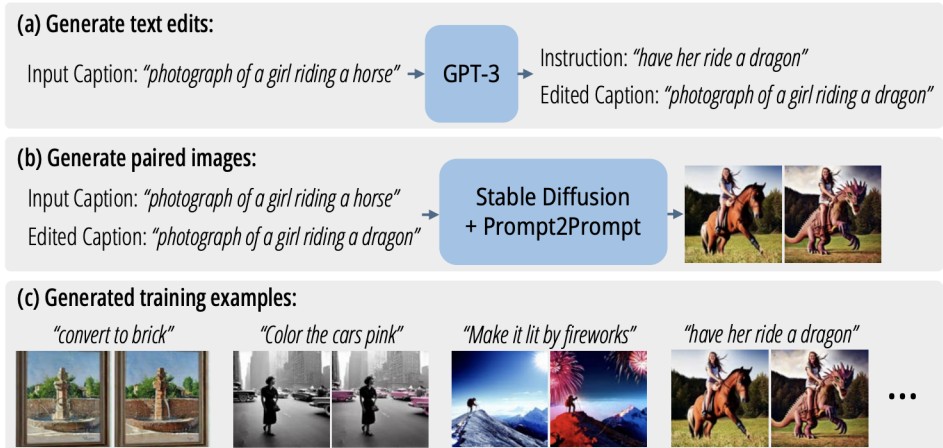

Figure 9: Data creation pipeline for Instruct-Pix2Pix (Brooks et al., 2023)

| Input | Instruction | Output | Explanation |
|---|---|---|---|
|  | Change the fashion to medieval |  | This is a good edit because the dress looks like something that could come out of medieval times ( armor) |
|  | Add street lamps |  | This edit is partially good because it adds the street lamps but it removes the benches from the picture |
|  | Make the hair of the boy spiky |  | This edit makes the hair more spiky but also changes the face. |
|  | Put carrots on the styrofoam tray |  | It put carrots on the styrofoam tray but it took off the other items from the tray which wasn't a command |

Table 5: Images from Indomain test set(Top two) and Out Of Domain test set (Bottom Two). The verdict chosen by human judges for the four transformations are i)Yes ii) Partially Yes iii)No iv) No

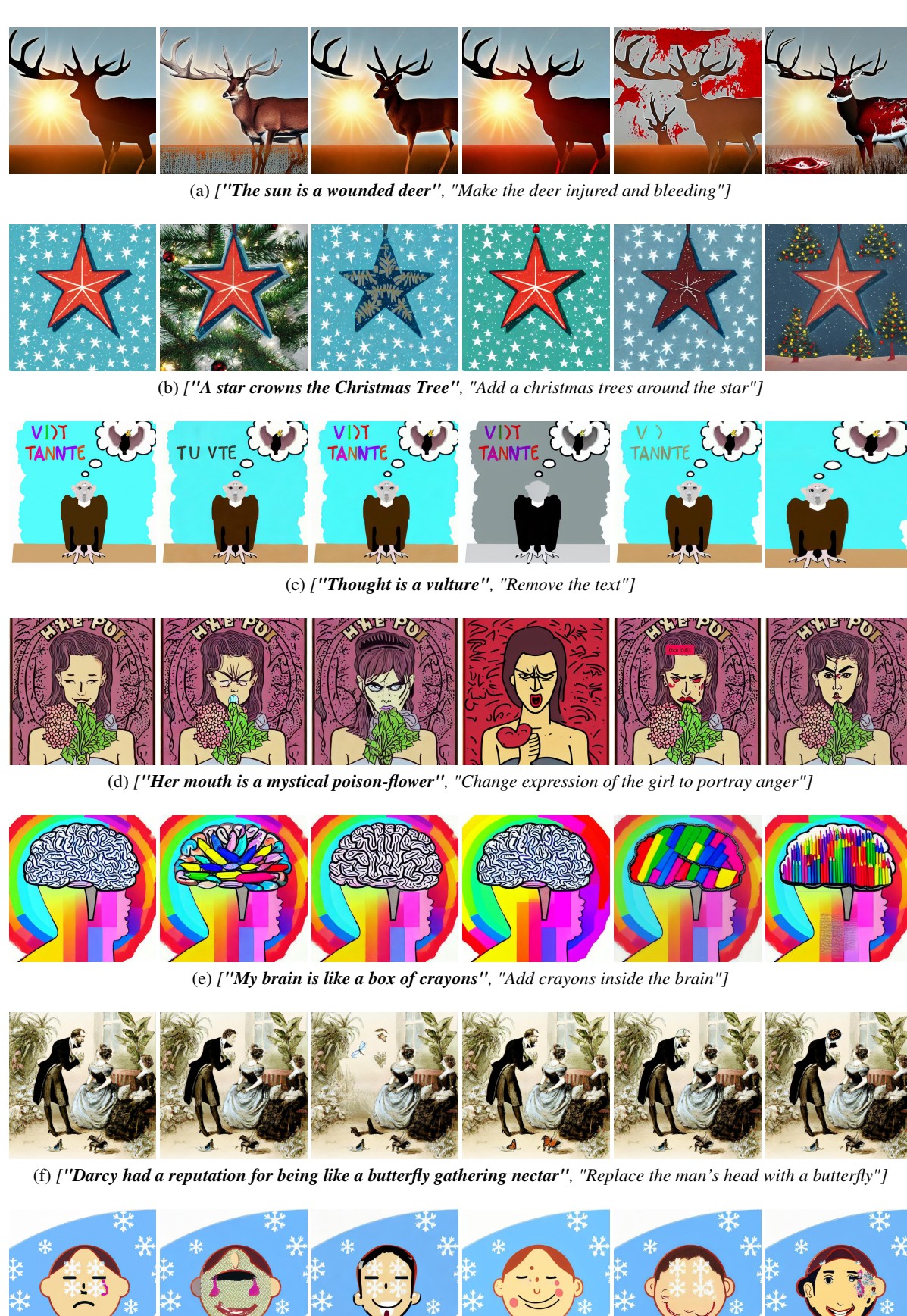

(a) ["**The sun is a wounded deer**", "Make the deer injured and bleeding"]

(b) ["**A star crowns the Christmas Tree**", "Add a christmas trees around the star"]

(c) ["**Thought is a vulture**", "Remove the text"]

(d) ["**Her mouth is a mystical poison-flower**", "Change expression of the girl to portray anger"]

(e) ["**My brain is like a box of crayons**", "Add crayons inside the brain"]

(f) ["**Darcy had a reputation for being like a butterfly gathering nectar**", "Replace the man's head with a butterfly"]

(g) ["**Everything is like a snowflake touching skin**", "Change expression of the boy to be happy"]

Figure 10: Sample image generations from the Metaphor dataset with associated metaphor-descriptions and corresponding edit instructions. The images displayed are following order (i) input-image, (ii) Grounded-Inpainting, (iii) X-Decoder, (iv) InstructPix2Pix, (v) InstructPix2Pix+BoundingBox, (vi) InstructPix2Pix+EntityMask

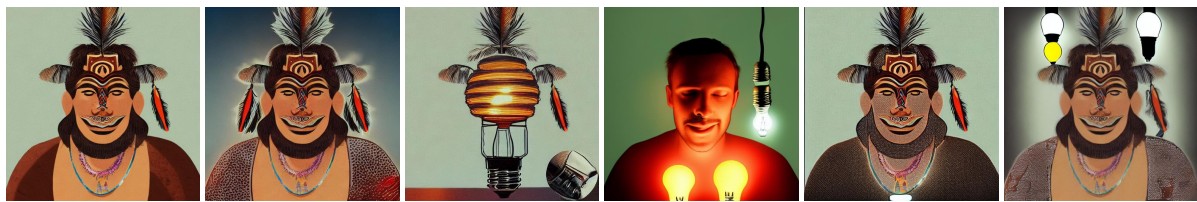

(a) *["He carried the knowledge or beliefs of his tribe", "Add glowing lightbulbs near the man's head"]*

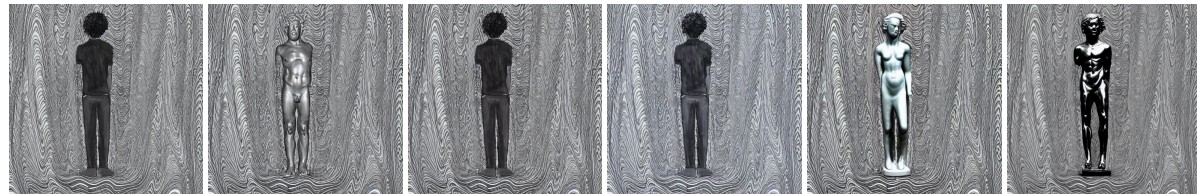

(b) *["He stands like a perfect marbled statue, never once looking away from me", "Turn the man into a marble statue looking forward"]*

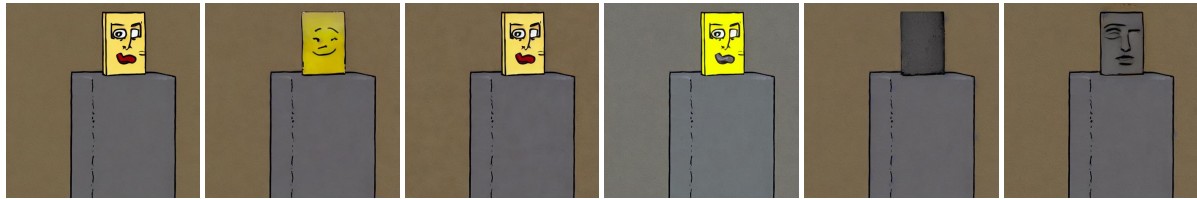

(c) *["He was like a block of cement", "Swap color of the head from yellow to grey"]*

Figure 11: Sample image generations from the Metaphor dataset with associated metaphor-descriptions and corresponding edit instructions. The images displayed are following order (i) input-image, (ii) Grounded-Inpainting, (iii) X-Decoder, (iv) InstructPix2Pix, (v) InstructPix2Pix+BoundingBox, (vi) InstructPix2Pix+EntityMask

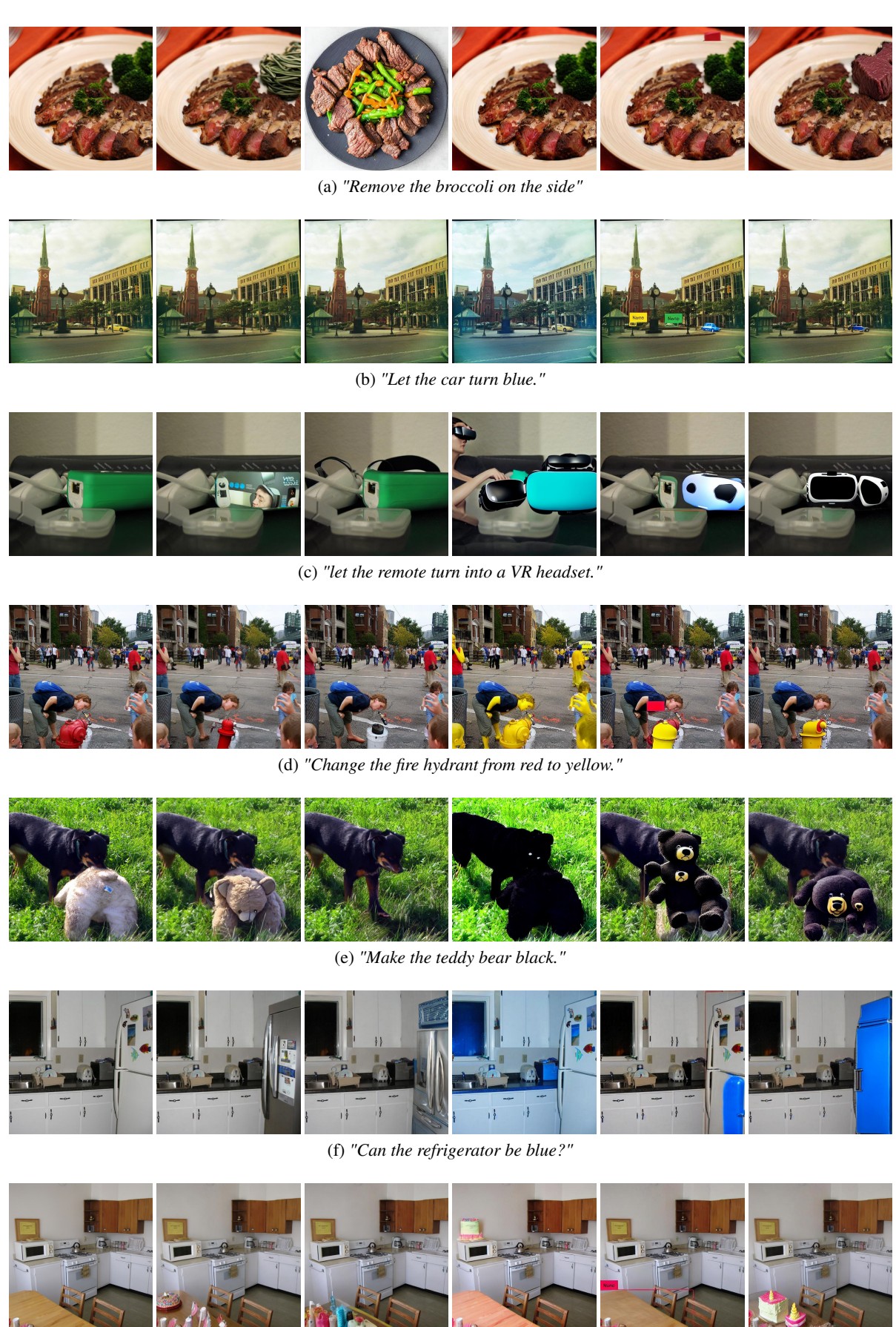

(a) *"Remove the broccoli on the side"*

(b) *"Let the car turn blue."*

(c) *"let the remote turn into a VR headset."*

(d) *"Change the fire hydrant from red to yellow."*

(e) *"Make the teddy bear black."*

(f) *"Can the refrigerator be blue?"*

(g) *"Have there be a birthday cake on the table."*

Figure 12: Sample generations for images from the Magicbrush dataset. The images displayed are following order (i) input-image, (ii) Grounded-Inpainting, (iii) X-Decoder, (iv) InstructPix2Pix, (v) InstructPix2Pix+BoundingBox, (vi) InstructPix2Pix+EntityMask

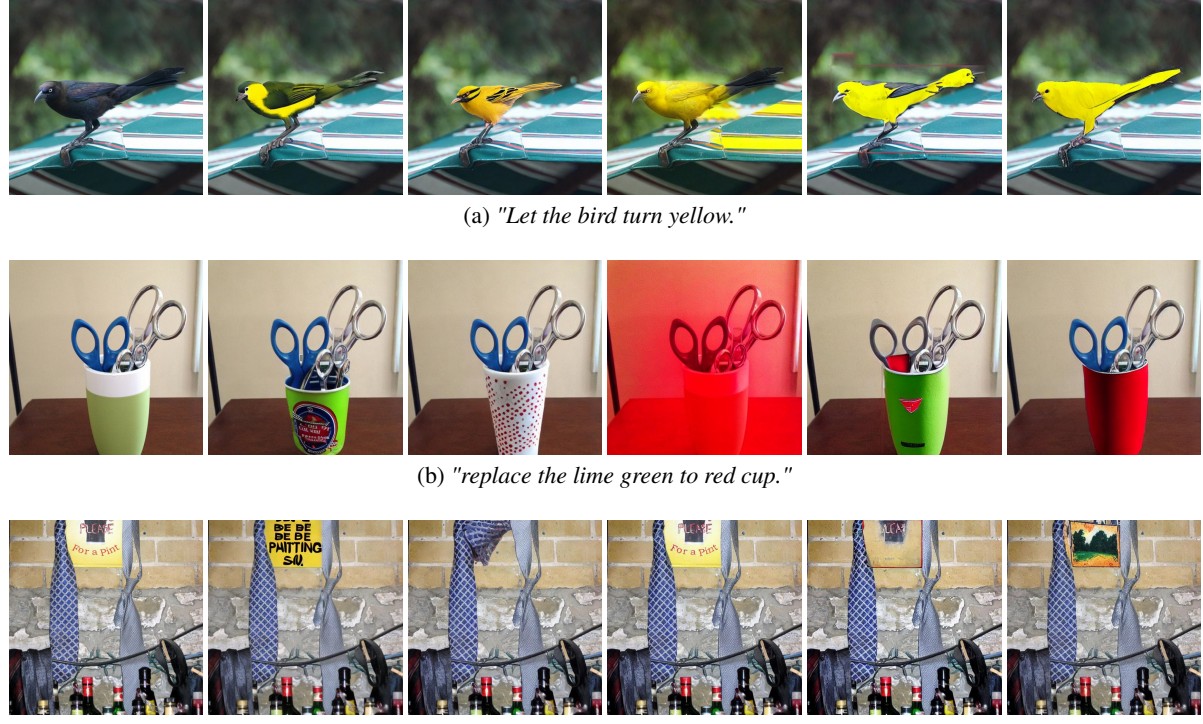

(a) *"Let the bird turn yellow."*

(b) *"replace the lime green to red cup."*

(c) *"let there be a painting instead of a sign."*

Figure 13: Sample generations for image outputs from the Magicbrush dataset. The images displayed are following order (i) input-image, (ii) Grounded-Inpainting, (iii) X-Decoder, (iv) InstructPix2Pix, (v) Instruct-Pix2Pix+BoundingBox, (vi) InstructPix2Pix+EntityMask