# OpenReview forum: "Learning to Follow Object-Centric Image Editing Instructions Faithfully"
_EMNLP/2023/Conference — EMNLP 2023 Findings_

### Official Review · Reviewer_Jm5D · 2023-08-05

**Typos Grammar Style And Presentation Improvements:** Please see Reasons to Reject.
**Soundness:** 4

**Excitement:**

3: Ambivalent: It has merits (e.g., it reports state-of-the-art results, the idea is nice), but there are key weaknesses (e.g., it describes incremental work), and it can significantly benefit from another round of revision. However, I won't object to accepting it if my co-reviewers champion it.

**Missing References:**

Please see Reasons to Reject.

**Paper Topic And Main Contributions:**

The paper addresses the challenges of underspecification, grounding and faithfulness in object-centric image editing with natural language instructions. The key idea of the paper is to enhance the quality of the paired data using recent advancements in segmentation, CoT prompting, and VQA. In addition, the work curates a test set consisting of both in-domain and out-of-domain examples. Furthermore, the paper proposes fine-tuning of the models with additional supervision signals such as bounding boxes and segmentation masks. Experiments demonstrate the utility of the proposed enhanced dataset and the fine-tuning strategies in improving automatic and human evaluation scores.

**Questions For The Authors:**

Please see my questions in Reasons to Accept and Reasons to Reject.

**Reasons To Accept:**

The proposed pipeline of enhancing paired data with CoT prompting, Grounding DINO, SAM,  BLIP-2 is really interesting. Are the authors planning to release the collected dataset? Evaluation with TIFA scores and human evaluation is solid and nicely demonstrates the impact of the proposed dataset.

**Reasons To Reject:**

I find the following major weaknesses of the work: First, it is not clear how the paper is leveraging the supervision signals (bboxes and segmentation masks) in the finetuning. Section 3 is really weak and does not provide any details on this important step. For example, do you pass them as additional channels into the UNet? Please provide more details as it seems to be one of the important contributions of this work. Next, in Figure 8, did you use the collected finetuned data for Instructpix2pix baseline? or Is the original data Instructpix2pix data used for both the baselines Instructpix2pix and InstructPix2Pix+Entity Mask. Please clarify. It would be nice to add more discussion/insights comparing the prior work on T2I results with visual metaphors datasets such as MetaCLUE (see citation below). There are other presentation issues in the paper. For example, the captions of figures show labels such as (a), (b), etc. But for most figures I couldn’t find them labeled in the figure which makes it super difficult to understand the inputs and outputs of models.

Akula, Arjun R., Brendan Driscoll, Pradyumna Narayana, Soravit Changpinyo, Zhiwei Jia, Suyash Damle, Garima Pruthi et al. "Metaclue: Towards comprehensive visual metaphors research." In Proceedings of the IEEE/CVF Conference on Computer Vision and Pattern Recognition, pp. 23201-23211. 2023.

**Reproducibility:**

2: Would be hard pressed to reproduce the results. The contribution depends on data that are simply not available outside the author's institution or consortium; not enough details are provided.

**Reviewer Confidence:**

4: Quite sure. I tried to check the important points carefully. It's unlikely, though conceivable, that I missed something that should affect my ratings.

---

> ### Author Rebuttal · Authors · 2023-08-28
>
> We thank the reviewer for their thoughtful feedback
>
> **First, it is not clear how the paper is leveraging the supervision signals (bboxes and segmentation masks) in the finetuning. Section 3 is really weak and does not provide any details on this important step. For example, do you pass them as additional channels into the UNet? Please provide more details as it seems to be one of the important contributions of this work.**
>
>
> We follow the same protocol as InstrPix2Pix training and use their codebase. For an image x, the diffusion process adds noise to the encoded latent $z = E(x)$ producing a noisy latent $z_{t}$ where the noise level increases over timesteps t ∈ T. We learn a network $\theta$ that predicts the noise added to the noisy latent  $z_{t}$  given image conditioning $c_{I}$ and text instruction conditioning $c_{T}$
> We minimize the latent diffusion objective and initialize the weights of  our model with a InstrPix2Pix checkpoint. To support image conditioning, we add additional input channels to the first convolutional layer, concatenating $z_{t}$  and $E(c_{I})$. All available weights of the diffusion model are initialized from the pre-trained checkpoints, and weights that operate on the newly added input channels are initialized to zero. We reuse the same text conditioning mechanism that was originally intended for captions to instead take as input the text edit instruction $c_{T}$
>
>
> We first create the improved and filtered dataset as shown in Figure 4. We then fine-tune 2 separate models
> * InstructPix2Pix+ Bounding Box
> * InstructPix2Pix+Entity Mask
>
> The training data for 1) has  image conditioning $c_{I}$ as an image with bounding box (Figure 5a) while training data for 2) has  image conditioning $c_{I}$  as an image with a segmentation mask on the desired entity/object.
> We will add these details in the final paper if accepted, given the additional one page.
>
> **Next, in Figure 8, did you use the collected finetuned data for Instructpix2pix baseline? or Is the original data Instructpix2pix data used for both the baselines Instructpix2pix and InstructPix2Pix+Entity Mask. Please clarify.**
>
> The original checkpoint provided by the InstructPix2Pix authors was used for the InstructPix2Pix baseline without any additional fine-tuning on our improved dataset. This checkpoint is finetuned with the newly refined image-instruction data pairs with two supervision signals as mentioned in the paper. For figure-8, the images used are a part of out-of-domain test-set from the Visual-Metaphor dataset (Chakrabarty et al) and we demonstrate the outputs from the InstructPix2Pix baseline and the InstructPix2Pix model fine-tuned with entity-mask.
>
> **It would be nice to add more discussion/insights comparing the prior work on T2I results with visual metaphors datasets such as MetaCLUE (see citation below).**
>
> We will cite MetaCLUE. It should be noted that the METACLUE paper’s data is not publicly released which makes any discussion or insight difficult. Finally we are not trying to generate or classify Visual metaphors but rather editing imperfect visual metaphors. The data from Chakrabarty et al (2023) was used as they had natural language instructions paired with imperfect visual metaphors and their data was publicly available.
>
> **There are other presentation issues in the paper. For example, the captions of figures show labels such as (a), (b), etc. But for most figures I couldn’t find them labeled in the figure which makes it super difficult to understand the inputs and outputs of models.**
>
> We will improve the presentation of the labels so that the input and output is more understandable.

---

### Official Review · Reviewer_1SU6 · 2023-08-05

**Soundness:** 4

**Excitement:**

3: Ambivalent: It has merits (e.g., it reports state-of-the-art results, the idea is nice), but there are key weaknesses (e.g., it describes incremental work), and it can significantly benefit from another round of revision. However, I won't object to accepting it if my co-reviewers champion it.

**Missing References:**

Couairon, Guillaume, et al. "Diffedit: Diffusion-based semantic image editing with mask guidance." arXiv preprint arXiv:2210.11427 (2022).

**Paper Topic And Main Contributions:**

The paper proposes a method for improving the training data for text-guided image editing by benefitting from recent advances in segmentation, chain of thought prompting, and VQA. By conducting a subjective evaluation and leveraging the TIFA score it is shown that by fine-tuning the InstrPix2Pix model on the enhanced data, its performance improves.

**Questions For The Authors:**

1- The authors should clearly mention and highlight their contributions.
2- Pls explain exactly how "Segment Anything Mask" is leveraged. What happens if the background needs to be fragmented and not consistent in a given image?

**Reasons To Accept:**

The paper proposes a procedure using the recent advances in computer vision and NLP domains to generate a reliable ground truth for text-guided image editing. Such a procedure can be used for generating more reliable supervised datasets in this field.

**Reasons To Reject:**

1- The resulting model is not compared against the SOTA models like Diffedit [1].
2- The qualitative results, considering the fact that there is no GT for more reliable quantitative analysis, are not enough.






[1] Couairon, Guillaume, et al. "DiffEdit: Diffusion-based semantic image editing with mask guidance." The Eleventh International Conference on Learning Representations. 2022.

**Reproducibility:**

4: Could mostly reproduce the results, but there may be some variation because of sample variance or minor variations in their interpretation of the protocol or method.

**Reviewer Confidence:**

3: Pretty sure, but there's a chance I missed something. Although I have a good feel for this area in general, I did not carefully check the paper's details, e.g., the math, experimental design, or novelty.

---

> ### Author Rebuttal · Authors · 2023-08-28
>
> We thank the reviewer for their thoughtful feedback
>
> **1- The resulting model is not compared against the SOTA models like Diffedit [1].**
> We will definitely cite it. We tried to compare against more recently published models and the idea of masking and inpainting is done in our Grounded Inpainting baseline (https://huggingface.co/runwayml/stable-diffusion-inpainting). We found some community implementations of DiffEdit but not the actual code from the authors which makes it harder to claim a fair comparison with the authors’ model . Additionally their data was not in the form of natural language instructions and hence not entirely comparable.
>
> **2- The qualitative results, considering the fact that there is no GT for more reliable quantitative analysis, are not enough.**
> In real world situations there is no ground truth so reference free quantitative evaluation is more reliable. We use TIFA scores for quantitative evaluation which is the state of art reference free metric accepted to ICCV 2023. TIFA correlates better than CLIPScore which is often used for text to image edit outputs. Finally,  for complex object centric edits, human evaluation is more trustworthy compared to any automatic evaluation. We performed human evaluation described in Section 5 lines 412-436.
>
> **Questions For The Authors:**
>
> **1- The authors should clearly mention and highlight their contributions.**
> Our contributions are explicitly stated in lines 123-139 in bullet points in introduction.  Our contributions are
> Improving the quality of existing paired datasets used for image editing via natural language instructions with the help of recent advances in segmentation, chain-of-thought prompting, and visual question answering.
> Curating a test set of diverse non-noisy instructions, consisting of both in-domain and out-of-domain examples and conducting a thorough evaluation across SOTA baselines and our model ablations.
> Demonstrating that fine-tuning a diffusion model on our parallel data enhanced with supervision signal leads to a significant improvement over several compelling baselines in terms of faithfulness using TIFA scores as well as instruction satisfiability. Our human evaluation corroborates these findings.
>
> **Pls explain exactly how "Segment Anything Mask" is leveraged. What happens if the background needs to be fragmented and not consistent in a given image?**
> Segment Anything Mask is used as an additional supervision signal during training and inference. During training, we provide the InstructPix2Pix model with an input image with a superimposed mask filled with a salt-and-pepper noise as shown in Fig-5(b), the edit instruction and the corresponding after image (generated using our data-refining pipeline). By providing a mask that highlights the specific regions of interest, the model gains a clearer understanding of where to apply the edits, ensuring that the modifications align with the user's intent and achieving faithfulness by ensuring that only the specified regions are transformed, leaving the rest of the image intact.
>
> For cases where the background needs to be fragmented such as instances with multiple objects of interest, the GroundingDINO module identifies multiple potential entities within the image, surpassing a predefined confidence threshold. Subsequently, SAM generates a segmentation mask that highlights all these predicted entities. These individual masks are merged to produce a final comprehensive mask thus accommodating instances that necessitate modifications over scattered elements in the image.

---

### Official Review · Reviewer_6zEB · 2023-08-09

**Soundness:** 4

**Excitement:**

3: Ambivalent: It has merits (e.g., it reports state-of-the-art results, the idea is nice), but there are key weaknesses (e.g., it describes incremental work), and it can significantly benefit from another round of revision. However, I won't object to accepting it if my co-reviewers champion it.

**Missing References:**

[1] arXiv:2304.06790 Inpaint Anything: Segment Anything Meets Image Inpainting


**Paper Topic And Main Contributions:**

This paper aims at a robust pipeline for instruction-based image editing. Starting from data preparation, they adopt external LLMs and chain-of-thought (CoT) reasoning to select feasible edit instructions as examples. Then, built upon segment anything model (SAM), they acquire the editing mask via visual grounding and perform image inpainting for the resulting image. This curated data can benefit the trained model and enhance the robustness of image editing.

**Questions For The Authors:**

Please see Reasons To Reject

**Reasons To Accept:**

+ This paper is well-written and easy to follow.
+ The proposed pipeline is elegant and training-free, which has the potential ability in adapting to various vision-and-language tasks.
+ They demonstrate superior performance from both quantitative and qualitative results, outperforming the baseline methods.

**Reasons To Reject:**

+ Despite being effective, the proposed pipeline seems to be a combination of existing modules. It is also similar to this concurrent work as SAM+Inpaining [1]. I am wondering if this achieves the EMNLP novelty bar.
+ Since the pipeline relies on SAM, it will be better to discuss how well SAM performs in this image editing case. If SAM fails, how to address and make the editing forward?
+ Some cases in image editing are not local manipulation (or not object-centric). For example, "brighten the whole photo" or "make it as cartoon". How to overcome this limitation in your pipeline?

**Reproducibility:**

4: Could mostly reproduce the results, but there may be some variation because of sample variance or minor variations in their interpretation of the protocol or method.

**Reviewer Confidence:**

5: Positive that my evaluation is correct. I read the paper very carefully and I am very familiar with related work.

---

> ### Author Rebuttal · Authors · 2023-08-28
>
> We thank the reviewer for their thoughtful feedback
>
> **Reason to reject 1**: Our novelty lies in identifying existing issues in one of the popular instructional edit papers, cleaning the noise, highlighting faithfulness issues and how to make it better. We will release the data, code and model, all of which are important contributions to the community. Finally, while our pipeline is a combination of existing modules it is in no way trivial and a successful execution leads to effective performance. Finally SAM+Inpaining [1] is published on April 13 on arxiv, is not peer reviewed and considered contemporaneous by ACL policy. In addition, the focus of our work is on following edit instructions, whereas SAM+Inpaininting requires explicitly specifying entities to be replaced and how to replace them.
>
> **Reason to reject 2:** Our training-data generation pipeline relies on the Grounding-DINO model to identify the entity and surround it with the bounding-box, which is then passed to SAM for generating a segmentation-mask. During this refining phase of training-data, we drop the instruction-image pairs if Grounding-DINO fails to detect the corresponding entity (not object-centric instructions) and for the cases where SAM partially (incorrectly) segments the entity, the image is discarded during the faithfulness (VQA) test. This 2-step check ensures that we curate the highest quality image-instruction pairs for our training data. During the inference phase, we carefully select object-centric instructions which ensure a bounding box around the entity from the Grounding-DINO module. SAM always generates a corresponding mask given a supervised set of points through background points/bounding boxes. SAM is highly efficient at segmenting the desired entity correctly and the performance of our pipeline is demonstrated through our evaluations on both in-domain and out-domain test-sets.
> However, in a human-AI collaborative environment, a user can always adjust the mask using additional foreground/background points as input to the SAM pipeline, to further enhance the edit quality.
>
>
> **Reason to reject 3:** Our paper focuses on object centric edits and faithfulness. Brighten the whole photo is a style transfer instruction which is already tackled by InstrPix2Pix. “Make it as a cartoon” can be handled by our pipeline. If “it” is a human or an object  that is mentioned in the caption  ChatGPT will detect the entity segment and the finetuned model will transform it to a cartoon.

---

### Official Review · Reviewer_QHP1 · 2023-08-12

**Soundness:** 3

**Excitement:**

3: Ambivalent: It has merits (e.g., it reports state-of-the-art results, the idea is nice), but there are key weaknesses (e.g., it describes incremental work), and it can significantly benefit from another round of revision. However, I won't object to accepting it if my co-reviewers champion it.

**Paper Topic And Main Contributions:**

The paper proposes a pipeline to improve the quality of language-guided image editing datasets. The authors also curate a test set consisting of both in-domain and out-of-domain instructions for benchmarking. They validate training mechanisms with box or mask supervision and show the improvement over previous editing models.

**Questions For The Authors:**

See above

**Reasons To Accept:**

- The proposed data processing pipeline potentially increases the quality of existing datasets, which can be quite noisy.
- The box and mask supervision demonstrates better performance than plain instruct-pix2pix.
- I think the data cleaning pipeline and test set would be beneficial to the community, provided that the authors do the reproduction work well afterwards.

**Reasons To Reject:**

- I am not fully convinced by the motivation behind the cleaning steps. While I do admit that some editing instructions can be quite challenging to achieve, the underspecification also allows flexibility and creativity for editing models. In addition, using GPT to determine if an instruction is feasible or not seems a bit unreliable to me as the generated captions can also be underspecified. Therefore, if the caption fails to describe all components clearly, GPT may tend to judge the instruction as impossible and thus remove high-quality, hard-positive editing image pairs from the dataset. Maybe I have missed something, can the authors justify this point?
- I think the box supervision is not a robust or plausible method given that Fig7 shows some generated artifacts, looking like an edge of a box or a red class label.
- The TIFA improvement is less significant than the human scores, which makes me wonder if the mask supervision only improves a small set of testing instructions given that they are trained on similar data samples or if it actually performs quite differently from instruct-pix2pix on many testing instructions but arrives at a similar TIFA score. Could the authors justify on the differences between TIFA and the human scores regarding the robustness of the former?

**Reproducibility:**

4: Could mostly reproduce the results, but there may be some variation because of sample variance or minor variations in their interpretation of the protocol or method.

**Reviewer Confidence:**

4: Quite sure. I tried to check the important points carefully. It's unlikely, though conceivable, that I missed something that should affect my ratings.

---

> ### Author Rebuttal · Authors · 2023-08-28
>
> We thank the reviewer for their thoughtful feedback
>
> **Reason to reject 1**: In object-centric edit we want the model to not change parts of the image that is not mentioned in the instruction. Our point is arguing for faithfulness and we think that faithfulness is compromised at the cost of creativity. When using GPT to determine if an instruction is feasible or not, 3 authors manually assessed more than 500 examples before going ahead and applying it on the entire data. There were errors in approximately 2% cases. In an ideal world every instruction needs to be verified by expert annotators but the data size is huge and we use ChatGPT to improve the InstrPix2Pix image which originally was automatically generated using GPT3 and was found to be very noisy. Finally, in the original InstrPix2Pix paper the edit instruction is generated on the original caption (see Fig 9 in Appendix) and so if the caption is underspecified and an instruction is nonsensical the issue is exacerbated. To ensure the quality of our training data, we opted to set aside samples with ambiguous captions. It is worth noting that there is no evidence to suggest that this approach omits high-quality, challenging editing image pairs from the initial InstrPix2Pix data. The authors of the original paper generated the data automatically, and didn't delve into the distribution of straightforward vs. challenging examples.
>
> **Reason to reject 2**: Yes there are artifacts in some images generated using box supervision, but not in all. A post processing step can easily get rid of these artifacts but since this was just an ablation we did not do anything further. Our best proposed model is InstrPix2Pix + Entity Mask which uses segmentation on top of the bounding box and produces the best result without any artifacts and the merit of our results should be based on that.
>
> **Reason to reject 3**:
> Human scores are harsher because they only choose between [0,0.5,1] for YES,PARTIALLY YES,NO. TIFA score for an individual sample will be the percentage of <questions,answer>  pairs generated by GPT3 where the answer matches correctly with the response of the VQA model. For example if GPT3 generated 10 questions,answers given a caption and the VQA model gets 7/10 answers correctly on the edited image the TIFA score is 0.7 and would always be higher as it is an approximate measure of the degree of faithfulness.TIFA scores while helpful as a metric requires VQA models to work reasonably well, which can be a limitation as sometimes they can fail to answer certain questions (line 459-461) so human scores are more reliable in this aspect

---

### Meta-Review · Area_Chair_aeWj · 2023-09-23

**Recommendation:** 3

**Metareview:**

The paper introduces a pipeline to enhance text-guided image editing by improving the quality of paired data through recent advancements in segmentation, chain-of-thoughts prompting, and VQA. While the proposed approach is intriguing and leverages important developments in computer vision and NLP, there are notable weaknesses in clarity and presentation. Specifically, the paper lacks detailed explanations regarding the use of supervision signals in fine-tuning and exhibits confusion regarding the treatment of fine-tuned data in experiments. Presentation issues, such as missing labels in figures, hinder the understanding of model inputs and outputs. Furthermore, the paper would benefit from a comparative analysis with other relevant datasets or methods, such as MetaCLUE. Despite these shortcomings, the use of TIFA scores and human evaluation adds credibility to the proposed dataset's impact, making it a valuable contribution to text-guided image editing with significant potential after addressing the mentioned issues.

---

### Decision · Program_Chairs · 2023-10-07

**Decision:**

Accept-Findings

**Comment:**

The paper introduces a pipeline to enhance text-guided image editing by improving the quality of paired data through recent advancements in segmentation, chain-of-thoughts prompting, and VQA. While the proposed approach is intriguing and leverages important developments in computer vision and NLP, there are notable weaknesses in clarity and presentation. Specifically, the paper lacks detailed explanations regarding the use of supervision signals in fine-tuning and exhibits confusion regarding the treatment of fine-tuned data in experiments. Presentation issues, such as missing labels in figures, hinder the understanding of model inputs and outputs. Furthermore, the paper would benefit from a comparative analysis with other relevant datasets or methods, such as MetaCLUE. Despite these shortcomings, the use of TIFA scores and human evaluation adds credibility to the proposed dataset's impact, making it a valuable contribution to text-guided image editing with significant potential after addressing the mentioned issues.